# Robotic Radical Nephroureterectomy with Bladder Cuff Excision for Upper Tract Urothelial Carcinoma: A Trend Analysis of Utilization and a Comparative Study

**DOI:** 10.3390/cancers14102497

**Published:** 2022-05-19

**Authors:** Hoyoung Bae, Jae Hoon Chung, Wan Song, Minyong Kang, Hwang Gyun Jeon, Byong Chang Jeong, Seong Il Seo, Seong Soo Jeon, Hyun Moo Lee, Hyun Hwan Sung

**Affiliations:** Department of Urology, Samsung Medical Center, Sungkyunkwan University School of Medicine, Seoul 06351, Korea; hoyoung.b@gmail.com (H.B.); jaehoontasker.chung@samsung.com (J.H.C.); wan.song@samsung.com (W.S.); m79.kang@samsung.com (M.K.); hwanggyun.jeon@samsung.com (H.G.J.); bc2.jung@samsung.com (B.C.J.); seongil.seo@samsung.com (S.I.S.); seongsoo.jeon@samsung.com (S.S.J.); hyunmoo.lee@samsung.com (H.M.L.)

**Keywords:** upper tract urothelial carcinoma, nephroureterectomy, robotic, trend

## Abstract

**Simple Summary:**

Robotic radical nephroureterectomy (RNU) for patients with upper tract urothelial carcinoma has only recently started to increase in Korea. The use of robotic RNU has been steadily increasing from 9% in 2017 to 67% in 2021 in our institution and replacing open and laparoscopic RNU. Perioperative outcomes including operation time, blood loss, and the length of hospital stay were not different between the robotic, open, and laparoscopic RNU groups. The 90-day complications did not differ significantly between the three groups. The three-year overall survival (OS) rates for open, laparoscopic, and robotic RNU were 91.8%, 90.4%, and 92.1%, respectively (*p* > 0.05). No differences in the intravesical recurrence-free survival, progression-free survival (PFS), cancer-specific survival (CSS), and overall survival (OS) were observed according to the surgical approach in the Kaplan–Meier survival analysis. Multivariate analysis showed that the surgical approach of RNU was not an independent predictor of PFS, CSS, and OS.

**Abstract:**

Purpose: To compare the perioperative outcomes and oncological results of open, laparoscopic, and robotic radical nephroureterectomy (RNU) in patients with upper tract urothelial carcinoma (UTUC) and to analyze trends in the utilization of RNU. Methods: From 2017 to 2020, the records of 61, 185, and 119 patients who underwent open, laparoscopic, and robotic RNU, respectively, were reviewed. Results: Baseline characteristics were not significantly different among the three groups. Robotic RNU has recently started to increase from 9% in 2017 to 67% in 2021. Operation time, blood loss, length of hospital stay, and 90-day complications were not different between the three groups. The three-year overall survival (OS) rates for open, laparoscopic, and robotic RNU were 91.8%, 90.4%, and 92.1%, respectively (*p* > 0.05). No differences in the progression-free survival (PFS), cancer-specific survival (CSS), and OS were observed according to the surgical approach in the Kaplan–Meier survival analysis. Multivariate analysis showed that surgical approach was not an independent predictor of PFS, CSS, and OS. Conclusion: The use of robotic RNU in patients with UTUC has been starting to increase and replace open and laparoscopic RNU. Perioperative outcomes, 90-day complications, and oncological outcomes of robotic RNU were not inferior to those of open and laparoscopic RNU.

## 1. Introduction

Urothelial carcinoma is the sixth most common tumor in developed countries, and upper urinary tract urothelial carcinoma (UTUC) is a relatively uncommon disease, accounting for 5–10% of urothelial carcinomas [1]. Open radical nephroureterectomy (RNU) with complete bladder cuff excision is the gold-standard for managing non-metastatic UTUC [2].

Since its introduction, laparoscopic RNU has emerged as an accepted minimally invasive alternative to open RNU [3]. There is a tendency towards equivalent oncological outcomes after laparoscopic or open RNU [4,5,6], although one small randomized study showed that laparoscopic RNU was inferior to open RNU for non-organ confined UTUC [7]. Laparoscopic RNU has been accepted as a safe modality by experienced surgeons when adhering to strict oncological principles [8].

Since the introduction of robot-assisted laparoscopic surgery, most open or laparoscopic prostatectomy and partial nephrectomy in the field of urologic oncology have been replaced by robotic surgery. A robotic RNU can also be considered a minimally invasive surgery (MIS) for treating UTUC [9,10,11], although there has been no prospective randomized study comparing robotic RNU and other modalities owing to the rarity of UTUC. Unlike the wide adoption of robotic surgery in prostate cancer or renal cell carcinoma, open or laparoscopic RNU is still the main treatment modality for UTUC; therefore, there are only a few studies comparing robotic RNU and other approaches.

This study aimed to compare perioperative surgical and oncological outcomes between open, laparoscopic, and robotic RNU in patients with non-metastatic UTUC and analyze the trends in the utilization of RNU.

## 2. Materials and Methods

### 2.1. Study Population

Robotic RNU has been adopted at our institution since 2017, and the records of patients who underwent RNU for UTUC from 2017 to 2020 were retrospectively reviewed. Patients who had undergone cystectomy before RNU and those with non-UTUC were excluded from the study. All RNUs were performed by seven experienced surgeons. Either the open, laparoscopic or robotic approach was selected based on the preference of the patient and operator. Annual trends in the utilization of robotic RNU were calculated from 2017 to 2021 at our institution and in Korea. The annual crude incidence of UTUC was obtained from the National Cancer Information Center in Korea (https://www.cancer.go.kr, accessed on 9 March 2022). UTUC was determined by C65 and C66 in the International Classification of Disease, 10th revision code, and the D code was not included in this data.

### 2.2. Surgical Technique of Open, Laparoscopic and Robotic RNU

Regardless of the surgical approach, the basic principle of RNU includes en bloc removal of the kidney, entire ureter, and bladder cuff. Two-incision approach (a flank incision for the dissection of the kidney and ureter and a Gibson incision for the bladder cuff excision) was performed for the open RNU. Pure laparoscopic RNU with transperitoneal or retroperitoneal approach was performed followed by an additional Gibson incision for the bladder cuff excision as performed in open RNU. All procedures of robotic RNU were performed using a five-port transperitoneal approach with the four-arm da Vinci Si^®^ or Xi^®^ Surgical Systems (Intuitive Surgical, Sunnyvale, CA, USA). The port placement for robotic RNU with the da Vinci Si^®^ or Xi^®^ system was followed as previously described by the other center using a single-dock approach without intraoperative patient repositioning [12,13,14]. The pneumoperitoneum pressure on laparoscopic and robotic RNU was maintained in the range of 12–14 mmHg. Lymph node dissection was performed in patients with suspected lymph node involvement on preoperative conventional imaging and some patients with high-risk disease.

### 2.3. Perioperative Surgical Outcomes

Baseline characteristics and perioperative data, such as the operation time, estimated blood loss (EBL), and length of postoperative hospital stay, were collected retrospectively. Charlson’s Comorbidity Index was calculated for each patient. The operation time was determined from incision to wound closure, including the docking time in the case of robotic RNU. Clavien-Dindo classification was used to report and grade post-operative complications within 90 days after surgery. Post-operative complications of grade 2 or higher and major complications (≥grade 3) were identified and classified into the open, laparoscopic, and robotic RNU groups.

### 2.4. Oncological Outcomes

Data on tumor stage, grade, location, multifocality, size, surgical margin, lymphovascular invasion (LVI), and adjuvant chemotherapy history were obtained from electronic medical records. Intravesical recurrence-free survival (IVRFS), progression-free survival (PFS), cancer-specific survival (CSS), and overall survival (OS) were calculated for the three groups.

### 2.5. Follow-Up

All patients were followed up for 3 months postoperatively for the first oncologic surveillance, including cystoscopy. Patients were followed up for 6 to 12 months using cystoscopy and imaging surveillance. All patients with pathologic tumor (pT) stage 3–4 and/or nodal disease were recommended for adjuvant chemotherapy and referred to the medical oncology department. Adjuvant chemotherapy was performed within three months after RNU, and the three courses of combined cisplatin or carboplatin plus gemcitabine was used every four weeks.

### 2.6. Statistical Analysis

Baseline demographic features, pathological variables and perioperative data were analyzed and reported using the mean with standard deviation for continuous variables, and the frequency and relative frequency for categorical variables. The chi-square test for categorical variables and one-way analysis of variance for continuous variables were used to compare differences between groups. All oncological survival rates were calculated using the Kaplan–Meier survival analyses with log-rank tests. A Cox proportional hazards regression model was used to determine the independent prognostic effects of different variables on survival. All statistical analyses were performed using IBM SPSS^®^ (version 27.0; SPSS Inc., Chicago, IL, USA) with significance defined as *p <* 0.05.

### 2.7. Ethical Statement

This study was approved by the Institutional Review Board (IRB) of Samsung Medical Center (No. 2022-03-105). The study followed the Declaration of Helsinki guidelines. The IRB waived the requirement for written informed consent owing to the retrospective design of the study.

## 3. Results

### 3.1. Baseline Characteristics

A total of 391 patients underwent RNU between 2017 and 2020 at our institution; 26 patients were excluded from the study because of cystectomy before RNU (*n =* 25) and non-UTUC pathology (*n =* 1). Among the 365 patients enrolled, 61, 185, and 119 patients underwent open, laparoscopic, and robotic RNU, respectively. As shown in Table 1, all baseline characteristics and pathological results of the robotic group were not significantly different from those of the open and laparoscopic approaches. However, the follow-up duration of the robotic RNU group was shorter than that of the open and laparoscopic groups (*p <* 0.001), although surgeries performed within the same period were investigated, as the two other approaches are being replaced by robotic surgery, which has been performed more recently. The overall follow-up duration was 27.8 ± 15.1 months. The tumor size in the open group was smaller than that in the laparoscopic RNU group. Pathological characteristics, including tumor size, multifocality, pT, pathologic node stage, grade, surgical margin status, and LVI rate, were not significantly different (*p* > 0.05). Lymphadenectomy was performed in 30.7% of all patients, and the implementation rate in the robotic group was not significantly different from the other two groups.

### 3.2. Perioperative Surgical Outcomes and 90-Day Complications Based on the Surgical Approach

Table 2 shows that the operation time, EBL, and postoperative hospital stay in the robotic group were similar to those in the open and laparoscopic groups (*p* > 0.05). Grade 2 or higher complications and major complications within 90 days of surgery were not significantly different among the groups (Table 2, *p* > 0.05). There were no 90-day mortality cases with any of the approaches. No differences in transfusion and 90-day re-admission were found between the groups (*p* > 0.05). There were no cases of conversion to the open approach in the robotic surgery group.

### 3.3. Oncological Outcomes

The Kaplan–Meier survival analyses with log-rank test revealed that no differences in IVRFS (*p =* 0.483), PFS (*p =* 0.842), CSS (*p =* 0.832), and OS (*p =* 0.819) were noted based on the surgical approach (Figure 1). The IVRFS in patients excluding a previous bladder tumor or concomitant bladder tumor was also not different between the three groups (*p* = 0.311). The 3-year PFS rates for open, laparoscopic, and robotic RNU were 77.1%, 74.2%, and 80.9%, respectively. Independent predictors of PFS in the multivariate analysis were ≥pT3, lymph node involvement, positive surgical margin, presence of LVI, tumor grade III, and no adjuvant chemotherapy (Table 3). In the multivariate analysis to predict CSS and OS, age, ≥ pT3, and LVI were independent factors (Table 3). The 3-year OS rates of open, laparoscopic, and robotic RNU were 91.8%, 90.4%, and 92.1%, respectively. Surgical approach was not an independent predictor of PFS, CSS, or OS in patients who underwent RNU for UTUC in the Cox-proportional hazards regression model (Table 3). In the subgroup analysis of advanced disease (≥pT3), MIS including laparoscopic and robotic surgery was not inferior to the open approach regarding IVRFS (*p =* 0.443), PFS (*p =* 0.738), CSS (*p =* 0.906) and OS (*p =* 0.994). The pattern of recurrence was not different according to the surgical approach. In all three groups, the most common primary sites of recurrence were regional lymph nodes, followed by the lung and liver. Unusual progression or metastasis such as peritoneal carcinomatosis was not observed in the robotic group or laparoscopic groups.

### 3.4. Annual Trends in the Utilization of Robotic RNU at Our Institution and in Korea

From 2017 to 2021, the number of robotic RNUs performed per year steadily increased from 9% to 67% at our institution (Figure 2, *p <* 0.001). Robotic RNU has replaced the laparoscopic approach, rather than open RNU, since 2017. Most recently, MIS including laparoscopic and robotic surgery has accounted for 90% of the total RNUs. The utilization of robotic RNU is also increasing in Korea, although the trends of annual crude incidence rates (per 100,000 persons) of UTUC are substantially steady (Figure 3).

## 4. Discussion

Randomized trials have compared robotic, laparoscopic, and open radical cystectomy in patients with muscle-invasive bladder cancer, showing that minimally invasive techniques, including robotic surgery, achieved equivalent oncological outcomes to the gold standard of open radical cystectomy [15,16]. The authors pointed out that the increased adoption of robotic surgery in clinical practice should lead to future trials to assess the true value of this approach in patients with other cancers [16]. However, it might still be challenging to plan a prospective randomized controlled study in the field of UTUC owing to its rarity; therefore, there were only a few level 1 studies on the management of UTUC [17,18,19]. The current retrospective study with intermediate follow-up demonstrated that the robotic approach of RNU was equivalent to laparoscopic and open RNU in terms of perioperative outcomes, 90-day complications, and oncological results, including IVRFS, PFS, CSS, and OS.

Despite the large difference in medical expenses according to the national medical insurance coverage in Korea, robotic approaches that are not covered by the national insurance is already becoming the main surgical method for radical prostatectomy in patients with prostate cancer or partial nephrectomy for renal cell carcinoma in tertiary cancer centers. Owing to the extra degrees of freedom, technically easier dissection, and short learning curve, the adoption of robotic surgery by surgeons is likely to be rapid. Recently, patients have had easy access to medical information through media or patient associations, and it seems that they aspire to undergo robotic surgery in anticipation of pain reduction and quick recovery after surgery, even though the medical cost of robotic surgery is much higher than that of the conventional approach. However, it is accepted that laparoscopic RNU is not technically challenging to access in patients with UTUC compared with laparoscopic surgeries in other types of malignancies; therefore, the adoption of robotic surgery in UTUC seems to be delayed for a while in Korea. Even in developed countries, robotic surgery for managing UTUC was not mainstream until mid-2010 [20]. They showed that from 2004 to 2013, MIS was increasingly used for managing UTUC, and by the end of the study, it accounted for approximately half of all RNUs performed in the USA. More recently, the number of laparoscopic RNUs performed has plateaued, whereas the number of robotic RNU continues increasing. The comfort and ergonomics of robotic RNU are considered to be superior to other methods, but these merits might be difficult to be shown with statistical significance. It was attempted to demonstrate the advantages of Robotic RNU indirectly through the analysis of the trend in the utilization of RNU surgery in the present study. The robotic surgery for UTUC has been rapidly increasing in Korea since 2016 and is mainly replacing laparoscopic surgery. Minimally invasive RNU has been the predominant method for treating UTUC at our institution. In recent years, the main surgical method of RNU has changed from laparoscopic surgery to a robotic approach.

A few studies suggest that the oncological outcomes of laparoscopic RNU may be poorer than those of open RNU, especially in patients with locally advanced high-risk UTUC [7,21]. A subgroup analysis of a prospective randomized study by Simone et al. showed that laparoscopic RNU is inferior to open surgery for pT3 and high-grade tumors of UTUC [7]. However, this study has several limitations, including the small sample size, single-center experience, personal preference for the laparoscopic technique, and failure to perform lymphadenectomies. The outcomes of RNU have not changed significantly over the past three decades, despite staging and surgical refinements [22]. Laparoscopic RNU is considered safe by experienced surgeons when adhering to strict oncological principles [4,23], although several precautions lowering the risk of tumor spillage during minimally invasive RNU are recommended. In the current study, the robotic approach was not inferior to open and laparoscopic surgeries in terms of oncological outcomes. RNU, based on the surgical approach, was not an independent predictor of PFS, CSS, or OS in the multivariate analysis. MIS, including laparoscopic and robotic RNU, was also not inferior to the open approach. In the subgroup analysis of locally advanced disease, there was no difference in the oncological outcomes between the three approaches. These results are consistent with the conclusions of other retrospective studies [9,10,11].

All types of surgery showed low perioperative morbidity and no mortality. Overall, the grade 2 or higher and major complication rate within 90 days after RNU were low at 18.1% and 2.2%, respectively. Postoperative hospital stay, EBL, transfusion, and readmission rate of robotic RNU were similar to those of the other two approaches. These results verified that robotic RNU is a safe and feasible surgical option for managing UTUC.

The robotic RNU technique has evolved since its introduction in 2006. Initial experience with robotic RNU required redocking of the robot or repositioning of the patient, which was time-consuming [24]. Subsequently, new and simplified approaches were introduced to eliminate the need for patient repositioning or robot redocking [14,25]. As our center has adopted robotic RNU relatively recently for managing UTUC, we followed robot docking only once from the beginning. All RNU surgeons were experienced in other robotic surgeries, such as robotic prostatectomy, partial nephrectomy, and radical cystectomy. The operation time of robotic RNU including docking time was not longer than that of the open and laparoscopic approaches.

To the best our knowledge, this is the first study to analyze the trends in RNU utilization in Korea. Unlike other robotic surgeries, the wide adoption of robotic RNU has been delayed. However, it was found that the number of robotic RNUs performed has rapidly increased in recent years even though the crude incidence was low and substantially similar over the years in Korea. Furthermore, the intermediate-term data showed equivalent perioperative complication profiles and survival outcomes among open, laparoscopic, and robotic RNUs for managing UTUC. Since data on robotic RNU for UTUC are scarce and prospective data are still lacking, this retrospective study should be considered meaningful in that additional evidence is accumulated.

Despite the strengths of the current study, several limitations should be noted. The retrospective nature of this single-center study and its non-randomized design with a surgical approach based on the surgeon’s and/or patient’s preference might contain significant selection bias. Second, the average follow-up period of the robotic group was shorter than that of the open and laparoscopic approaches because of the transition period in which the surgical methods are changing. The intermediate follow-up duration of an average of 27.9 months was also relatively short for inferring oncological outcomes. However, various prognostic factors to predict oncological results could be identified in the multivariate analysis, and there were no restrictions in confirming perioperative surgical outcomes and 90-day complications in this study. Third, a significant portion of patients (69%) did not undergo lymphadenectomy, implying a bias of understaging. Template-based and complete lymphadenectomy improves CSS and should be offered to all patients scheduled for RNU for high-stage (≥pT2) or high-risk non-metastatic UTUC [26]. Lastly, at the time of this study, data on the annual crude incidence of UTUC from the National Cancer Information Center in Korea were only available until 2019. However, it is expected that there will be no significant changes thereafter.

## 5. Conclusions

The wide adoption of robotic RNU in patients with UTUC seems to have been delayed, although its use has been rapidly increasing in Korea. Robotic RNU is mainly replacing laparoscopic surgery, and MIS has been the predominant treatment method for UTUC at our institution. Robotic RNU, as an MIS, is a safe and feasible strategy with low morbidity and no mortality. At the intermediate follow-up, the survival outcomes of robotic RNU were equivalent to those of the open and laparoscopic approaches. Further prospective studies are required to verify these results.

## Figures and Tables

**Figure 1 cancers-14-02497-f001:**
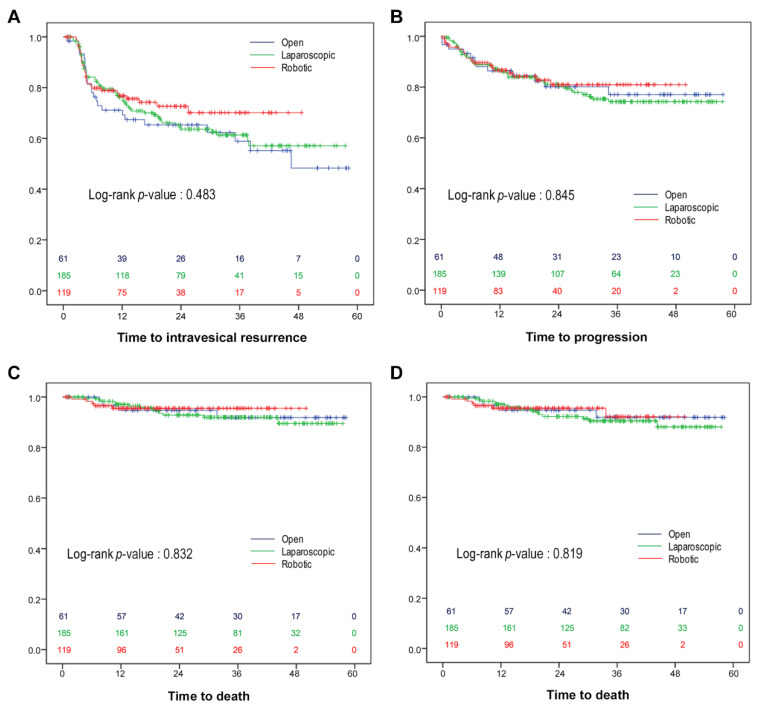
Kaplan–Meier survival analyses estimated intravesical recurrence-free survival (**A**), progression-free survival (**B**), cancer-specific survival (**C**), and overall survival (**D**), stratified by the surgical approach. Red, robotic; green, laparoscopic; blue, open radical nephroureterectomy.

**Figure 2 cancers-14-02497-f002:**
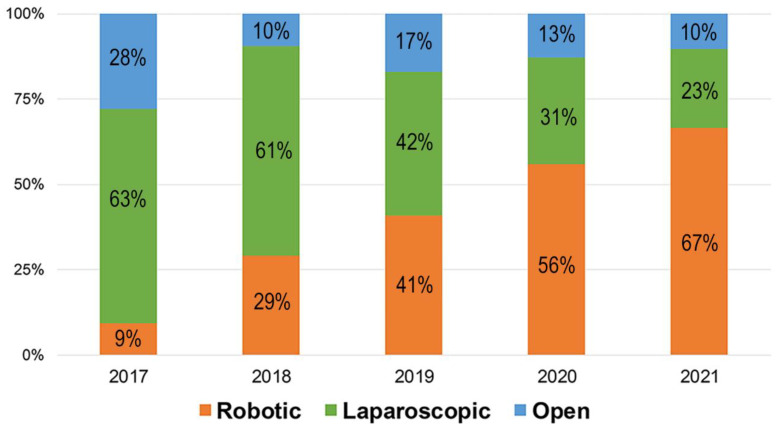
Proportion of radical nephroureterectomy stratified by the surgical approach.

**Figure 3 cancers-14-02497-f003:**
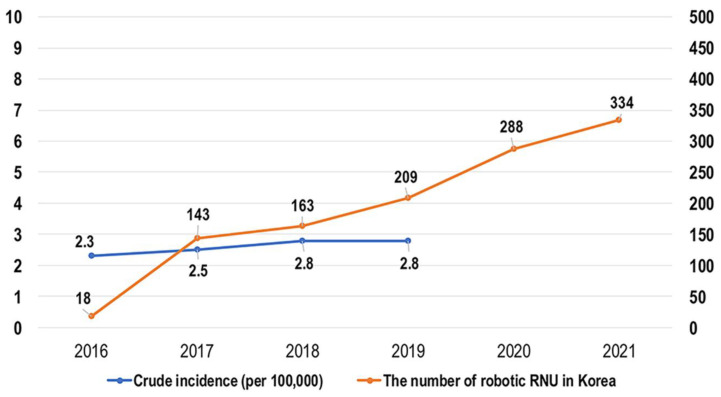
The utilization of robotic radical nephroureterectomy and annual crude incidence rate (per 100,000 persons) of upper tract urothelial carcinoma in Korea. RNU, radical nephroureterectomy.

**Table 1 cancers-14-02497-t001:** Baseline characteristics.

*N* = 365	Open (*n* = 61)	Laparoscopic (*n* = 185)	Robotic (*n* = 119)	*p*-Value
Age, years	69.7 ± 9.4	67.6 ± 9.6	68.5 ± 9.1	0.295
Sex, male, % (n)	67.2% (41)	70.8% (131)	71.4% (85)	0.830
BMI, kg/m^2^	25.3 ± 3.2	24.6 ± 3.2	25.2 ± 3.8	0.171
F/U duration, months	32.4 ± 16.4	29.92 ± 15.3	22.0 ± 12.4	<0.001
Year of surgery, % (n)				<0.001
2017	27.8% (27)	62.9% (61)	9.3% (9)	
2018	9.4% (10)	61.3% (65)	29.2% (31)	
2019	17.1% (13)	42.1% (32)	40.8% (31)	
2020	12.8% (11)	31.4% (27)	55.8% (48)	
CCI, grade ≥ 3, % (n)	59.0% (36)	64.9% (120)	58.0% (69)	0.435
Tumor location, % (n)				0.863
Ureter	52.5% (32)	49.7% (92)	45.4% (54)	
Renal pelvis	41.0% (25)	42.2% (78)	44.5% (53)	
Both	6.6% (4)	8.1% (15)	10.1% (12)	
Multifocality, % (n)	8.2% (5)	17.8% (33)	20.2% (24)	0.117
Tumor size, cm	3.0 ± 1.4	3.9 ± 2.8	3.6 ± 2.0	0.047
pT stage, pTa, % (n)	9.8% (6)	11.4% (21)	8.4% (10)	0.742
pT1	37.7% (23)	36.8% (68)	31.9% (38)	
pT2	19.7% (12)	18.4% (34)	13.4% (16)	
pT3-4	32.8% (20)	32.4% (60)	44.5% (53)	
pTis	0	1.1% (2)	1.7% (2)	
pN stage, pN0, % (n)	19.7% (12)	22.7% (42)	23.5% (28)	0.977
pN+	8.2% (5)	8.6% (16)	7.6% (9)	
pNx	72.1% (44)	68.6% (127)	68.9% (82)	
Tumor grade, I, % (n)	3.3% (2)	4.3% (8)	3.4% (4)	0.222
II	49.2% (30)	52.4% (97)	39.5% (47)	
III	47.5% (29)	43.2% (80)	57.1% (68)	
Positive surgical margin, % (n)	3.3% (2)	4.9% (9)	1.7% (2)	0.340
Concomitant CIS, % (n)	4.9% (3)	8.6% (16)	9.2% (11)	0.579
LVI, % (n)	11.5% (7)	15.7% (29)	21.8% (26)	0.171
Adjuvant chemotherapy, % (n)	19.7% (12)	22.2% (41)	31.1% (37)	0.129

BMI, body mass index; CCI, Charlson Comorbidity Index; CIS, carcinoma in situ; F/U, follow-up; LVI, lymphovascular invasion; pN, pathologic node; pT, pathologic tumor.

**Table 2 cancers-14-02497-t002:** Hospital course and complications within 90 days after surgery.

*N* = 372	Open (*n* = 61)	Laparoscopic (*n* = 185)	Robotic (*n* = 119)	*p*-Value
Operative time, min	211 ± 63	196 ± 74	189 ± 54	0.101
EBL, mL	161 ± 84	157 ± 123	147 ± 113	0.676
Hospital stay, days	7.5 ± 1.9	7.2 ± 2.4	7.3 ± 3.4	0.754
90-day complications				
Transfusion, % (n)	4.9 (3)	6.5 (12)	5.9 (7)	0.902
Re-admission, % (n)	0	1.6 (3)	3.4 (4)	0.273
≥Grade 2, % (n)	16.4% (10)	16.8% (31)	21.0% (25)	0.599
≥Grade 3, % (n)	1.6% (1)	1.6% (3)	3.4% (4)	0.569

EBL, estimated blood loss.

**Table 3 cancers-14-02497-t003:** Multivariate analysis with Cox-proportional hazards regression to predict progression-free survival, cancer-specific survival, and overall survival.

	Progression-Free Survival ^a)^		Cancer-Specific Survival ^b)^		Overall Survival ^b)^	
	HR (95% CI)	*p*-Value	HR (95% CI)	*p*-Value	HR (95% CI)	*p*-Value
Age			1.08 (1.02–1.14)	0.012	1.06 (1.01–1.12)	0.022
pT staging, ≥T3	3.58 (1.83–7.03)	<0.001	4.14 (1.16–14.69)	0.028	4.04 (1.28–12.82)	0.018
pN staging, positive	3.80 (1.82–7.92)	<0.001				
Positive surgical margin	2.64 (1.01–6.93)	0.048				
Presence of LVI	4.2 (2.34–7.52)	<0.001	6.62 (2.06–21.24)	0.002	8.88 (3.01–26.24)	<0.001
Tumor grade III	1.95 (1.05–3.62)	0.034				
Adjuvant chemotherapy	0.35 (0.17–0.70)	0.003	0.35 (0.11–1.13)	0.080	0.25 (0.08–0.78)	0.017
Surgical approach		0.580		0.970		0.699
Robotic	reference		reference		reference	
Open	1.29 (0.71–2.33)		1.12 (0.35–3.54)		1.28 (0.45–3.61)	
Laparoscopic	1.45 (0.68–3.11)		0.98 (0.22–4.40)		0.77 (0.18–3.30)	

CI, confidential interval; CIS, carcinoma in situ; HR, hazard ratio; LVI, lymphovascular invasion. ^a)^ Adjusted with age, sex, Charlson comorbidity index, tumor size, tumor location, presence of carcinoma in situ. ^b)^ Adjusted with sex, Charlson comorbidity index, tumor size, tumor location, tumor grade, pN staging, margin status, presence of carcinoma in situ.

## Data Availability

The data presented in this study are available on request from the corresponding author.

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
