# Peer review of "Robotic Radical Nephroureterectomy with Bladder Cuff Excision for Upper Tract Urothelial Carcinoma: A Trend Analysis of Utilization and a Comparative Study"

_cancers, 2022, doi:10.3390/cancers14102497_

Round 1
Reviewer 1 Report
In the present study, authors compared perioperative surgical and oncological outcomes between open, laparoscopic, and robotic radical nephroureterectomy in patients with non-metastatic upper urinary tract urothelial carcinoma and analyzed the trends in the utilization of radical nephroureterectomy. They found that perioperative surgical and oncological outcomes were not different among the three groups and the utilization of robotic radical nephroureterectomy is increasing in Korea. The manuscript was well-written, but further information is needed.
Comments:
- The surgeon can vary the surgical approach according to the clinical stage. Therefore, the author should present the clinical stage (T, N) according to the surgical method.
- I wonder if robot surgery is a factor associated with intravesical recurrence. In a multicenter study involving your institute, the surgical approach was revealed as an independent prognostic factor for intravesical recurrence. (Cancer Res Treat. 2019;51(1):240-251.)
- I wonder if a history of previous bladder tumor and presence of concomitant bladder tumor are factors associated with intravesical recurrence, progression, cancer-specific death, and all-cause death. In a multicenter study involving your institute, they were revealed as independent prognostic factors for intravesical recurrence, progression, cancer-specific death, and all-cause death. (Cancer Res Treat. 2019;51(1):240-251.)
- I wonder if diagnostic ureteroscopy performed prior to radical nephroureterectomy affects intravesical recurrence. In research previously published by your institution, diagnostic ureteroscopy for upper urinary tract urothelial carcinoma increased intravesical recurrence rate after radical nephroureterectomy. (PLoS One. 2015;10(11):e0139976. & Front Oncol. 2021;11:730114.)
- I wonder if the pattern of recurrence is different depending on the surgical approach. Peritoneal carcinomatosis has been reported only in minimally invasive surgery.
- The surgical techniques of open and laparoscopic radical nephroureterectomy should also be described in detail. It should be stated whether the laparoscopic radical nephroureterectomy was hand-assisted or pure laparoscopic. In addition, it should be described whether the bladder cuff excision was performed while maintaining pneumoperitoneum or was performed openly through an incision.

Author Response
Dear reviewer
I thank the reviewers of the “Cancers” for taking their time to review my article. I have made some corrections and clarifications in the manuscript after going over the reviewer’s comments. Included are answers to reviewer’s question. The answers are attached as a PDF file.
I hope the revised manuscript will better meet the requirements of the “Cancers” for publication. I thank you again for the constructive review by the reviewers.
Sincerely yours
Hyun Hwan Sung, M.D. & Ph.D.

Reviewer 2 Report
Thank you very much for giving me this opportunity to review the article entitled "Robotic radical nephroureterectomy with bladder cuff excision for upper tract urothelial carcinoma: A trend analysis of utilization and a comparative study." In this article, the authors retrospectively analyzed their clinical data regarding radical nephroureterectomy in order to reveal the benefit of robot-assisted surgery. They reported no significant differences between robotic and other surgeries in terms of operative time, blood loss, or perioperative complications. The focus of this study is potentially interesting and worthy of eventual publication. However, some modifications are required. The followings are my comments and questions. (Major) -What is the benefit of robotic surgery? The authors reported no significant differences between robotic surgery and other surgeries. It seems that this content merely states the non-inferiority of robotic surgery. It would be better to describe the statistically significant points regarding the benefit of robotic surgery. In radical nephroureterectomy, robotic surgery is unlikely to be more noninvasive than laparoscopic surgery. The same is true about surgical costs. (Minor) In Simple summary, the ratio of robotic RNU was written as "9% in 2017 to 67% in 2021." This description seems to be hard to understand the data source. Please state the data source, whether it is from the authors' institution, Korea, or worldwide.Author Response
Dear reviewer
I thank the reviewers of the “Cancers” for taking their time to review my article. I have made some corrections and clarifications in the manuscript after going over the reviewer’s comments. Included are answers to reviewer’s question. The answers are as follows.
Thank you very much for giving me this opportunity to review the article entitled "Robotic radical nephroureterectomy with bladder cuff excision for upper tract urothelial carcinoma: A trend analysis of utilization and a comparative study." In this article, the authors retrospectively analyzed their clinical data regarding radical nephroureterectomy in order to reveal the benefit of robot-assisted surgery. They reported no significant differences between robotic and other surgeries in terms of operative time, blood loss, or perioperative complications. The focus of this study is potentially interesting and worthy of eventual publication. However, some modifications are required. The followings are my comments and questions.
(Major) -What is the benefit of robotic surgery? The authors reported no significant differences between robotic surgery and other surgeries. It seems that this content merely states the non-inferiority of robotic surgery. It would be better to describe the statistically significant points regarding the benefit of robotic surgery. In radical nephroureterectomy, robotic surgery is unlikely to be more noninvasive than laparoscopic surgery. The same is true about surgical costs.
-This is a very important question. As we all know, pure laparoscopic nephrectomy is not relatively difficult to perform, but it could be difficult to perform pure laparoscopic nephroureterectomy up to bladder cuff excision. Therefore, even with laparoscopic RNU, an additional incision is often performed for bladder cuff excision. There is no statistical difference in the results of these three procedures in various aspects in the current study as you stated. But, the comfort and ergonomics of robotic RNU are considered to be superior to other methods, and these merits might be difficult to be shown statistically. It was attempted to demonstrate the advantages of Robotic RNU indirectly through the analysis of the trend of utilization of surgery in the present study.
These comments were added in the discussion section.
(Minor) In Simple summary, the ratio of robotic RNU was written as "9% in 2017 to 67% in 2021." This description seems to be hard to understand the data source. Please state the data source, whether it is from the authors' institution, Korea, or worldwide.
-Thanks for your comments. That sentence was revised and clarified as your recommend.
I hope the revised manuscript will better meet the requirements of the “Cancers” for publication. I thank you again for the constructive review by the reviewers.
Sincerely yours
Hyun Hwan Sung, M.D. & Ph.D.
Reviewer 3 Report
The manuscript clearly revealed the robotic nephroureterectomy is not inferior to open or laparoscopic nephroureterectomy among the Korean Cohort.
Major point
- Some past studies had shown that pneumoperitoneum time is associated with IVR. The authors should at least demonstrate the pneumoperitoneum pressure on laparoscopic and robotic nephroureterectomy.
- The methods of bladder cuff resection might impact on IVR after surgery. The authors should demonstrate the methods of bladder cuff resection on open and laparoscopic nephroureterectomy.
- Some past studies had shown that preoperative positive urine cytology is associated with IVR. The authors should demonstrate the results of urine cytology if possible.
- Authors should demonstrate the indication and regimens of adjuvant chemotherapy, because about 40% of patients treated with non-open nephroureterectomy underwent adjuvant chemotherapy and no adjuvant chemotherapy was the independent risk factor for PFS and OS.
Minor point
- L 178  In the univariate analyses,→The Kaplan-Meier survival analyses with log-rank test revealed that
- It should be clearly demonstrated that Table 3 show multivariate analyses.
Author Response
Dear reviewer
I thank the reviewers of the “Cancers” for taking their time to review my article. I have made some corrections and clarifications in the manuscript after going over the reviewer’s comments. Included are answers to reviewer’s question. The answers are as follows.
Major point
- Some past studies had shown that pneumoperitoneum time is associated with IVR. The authors should at least demonstrate the pneumoperitoneum pressure on laparoscopic and robotic nephroureterectomy.
- Thanks for your comments. Unfortunately, we do not have the data of pneumoperitoneum time. But, the pneumoperitoneum pressure on laparoscopic and robotic RNU was maintained in the range of 12-14 mmHg in our institution and this sentence was added in the method section.
- The methods of bladder cuff resection might impact on IVR after surgery. The authors should demonstrate the methods of bladder cuff resection on open and laparoscopic nephroureterectomy.
- We totally agree with your opinion. Additional comments were described in the method section as your recommend.
- Some past studies had shown that preoperative positive urine cytology is associated with IVR. The authors should demonstrate the results of urine cytology if possible.
- It is an important query. Unfortunately, at the time of study, we did not include the data of preoperative urine cytology. As your recommend, we will include data of preoperative urine cytology in the next study.
- Authors should demonstrate the indication and regimens of adjuvant chemotherapy, because about 40% of patients treated with non-open nephroureterectomy underwent adjuvant chemotherapy and no adjuvant chemotherapy was the independent risk factor for PFS and OS.
- Thanks for your kind remarks. Indication and regimens of adjuvant chemotherapy were described additionally in the method section.
Minor point
- L 178 in the univariate analyses,→The Kaplan-Meier survival analyses with log-rank test revealed that
- Thanks for your remark, and that sentence was revised.
- It should be clearly demonstrated that Table 3 show multivariate analyses.
-Table 3 was revised.
I hope the revised manuscript will better meet the requirements of the “Cancers” for publication. I thank you again for the constructive review by the reviewers.
Sincerely yours
Hyun Hwan Sung, M.D. & Ph.D.
Round 2
Reviewer 1 Report
I thank the Authors for having addressed all the points raised. I think that the revised manuscript is now fully suitable for publication.
Reviewer 3 Report
The manuscript has been corrected appropriately.